# Co-Processed Olive Oils with *Thymus mastichina* L.—New Product Optimization

**DOI:** 10.3390/life11101048

**Published:** 2021-10-06

**Authors:** Fátima Peres, Marta Roldão, Miguel Mourato, Luisa L. Martins, Suzana Ferreira-Dias

**Affiliations:** 1Instituto Politécnico de Castelo Branco, Escola Superior Agrária, 6000-909 Castelo Branco, Portugal; fperes@ipcb.pt; 2LEAF, Linking Landscape, Environment, Agriculture and Food, Instituto Superior de Agronomia, Universidade de Lisboa, 1349-017 Lisbon, Portugal; marta_roldao97@hotmail.com (M.R.); mmourato@isa.ulisboa.pt (M.M.); luisalouro@isa.ulisboa.pt (L.L.M.)

**Keywords:** co-extraction, flavored oil, response surface methodology, phenols, thyme

## Abstract

Olive co-processing consists of the addition of ingredients either in the mill or in the malaxator. This technique allows selecting the type of olives, the ingredients with the greatest flavoring and bioactive potential, and the technological extraction conditions. A new product—a gourmet flavored oil—was developed by co-processing olives with *Thymus mastichina* L. The trials were performed using overripe fruits with low aroma potential (cv. ‘Galega Vulgar’; ripening index 6.4). Experimental conditions were dictated by a central composite rotatable design (CCRD) as a function of thyme (0.4−4.6%, *w/w*) and water (8.3−19.7%, *w/w*) contents used in malaxation. A flavored oil was also obtained by adding 2.5% thyme during milling, followed by 14% water addition in the malaxator (central point conditions of CCRD). The chemical characterization of the raw materials, as well as the analysis of the flavored and unflavored oils, were performed (chemical quality criteria, sensory analysis, major fatty acid composition, and phenolic compounds). Considering chemical quality criteria, the flavored oils have the characteristics of “Virgin Olive Oil” (VOO), but they cannot have this classification due to legislation issues. Flavored oils obtained under optimized co-processing conditions (thyme concentrations > 3.5−4.0% and water contents varying from 14 to 18%) presented higher phenolic contents and biologic value than the non-flavored VOO. In flavored oils, thyme flavor was detected with high intensity, while the defect of “wet wood”, perceived in VOO, was not detected. The flavored oil, obtained by *T. mastichina* addition in the mill, showed higher oxidative stability (19.03 h) than the VOO and the co-processed oil with thyme addition in the malaxator (14.07 h), even after six-month storage in the dark (16.6 vs. 10.3 h).

## 1. Introduction

Virgin olive oil is the oil extracted from the fruits of the olive tree (*Olea europaea* L.) using exclusively mechanical extraction techniques under conditions that will not affect the original composition of the oil. In the last decades, many research studies have shown that virgin olive oil has bioactive properties with impacts on health, particularly in preventing cardiovascular diseases, cancer, diabetes, and neurodegenerative diseases [1,2,3,4]. These benefits are due not only to its fatty acid profile, especially to the high content in oleic acid, but also to the presence of several bioactive compounds [1,5,6,7]. This evidence led the European Union, in 2006, to approve nutritional and health claims for virgin olive oil (VOO), to be included on the label. They concern monounsaturated fatty acids, oleic acid, and unsaturated acids, vitamin E, and more recently, in 2012, polyphenols in the olive oil [8,9].

Most of the Portuguese olive orchards in organic production are predominantly rain-fed and very much based on autochthonous varieties, which are characterized by unique sensory properties. Combining these aspects with the consumers’ trend towards products offering health benefits, the great challenge to promote and protect this type of olive orchard is related to the production of high-quality oils (extra virgin olive oil—EVOO) and novel added-value products. Labeling these products with nutrition and health claims is very important in terms of consumers’ acceptance and choice. In recent years, innovation in olive oils has included flavoring with different ingredients, with the aim of improving their sensory and nutritional properties, as well as shelf life. Thus, this type of product is highly appreciated, especially among consumers outside the Mediterranean countries, reaching values above EUR 50 per liter. The added compounds may have several health benefits due to the presence of natural bioactive substances with antioxidant and/or antimicrobial properties and may contribute to increasing olive oil’s resistance to oxidation [10]. Vegetables, aromatic herbs, fruits, nuts, essential oils, and spices are the most common ingredients used, added either as an infusion, ethanolic extracts of essential oils, or by co-processing [10,11,12,13,14,15,16].

Co-processing, also known as co-extraction, is an alternative method to infusion techniques, or to the addition of ethanolic extracts, to obtain flavored or enriched oils. It consists of the addition of ingredients, for example, fruits or aromatic plants, during milling or in the malaxation step of the olive oil extraction process. This technique allows for selecting both the type of olives (e.g., cultivar and ripening stage), the ingredient(s) with the greatest flavoring and/or bioactive potential, as well as the extraction conditions. Moreover, this method does not need the filtration step, conversely to when infusion is performed. For olive oils from organic farming, the production of co-processed oils can be an opportunity for differentiation, creating products mainly for the non-traditional consumer market. Although studies of co-processing are scarce, some show that using co-extraction with the addition of lemon, bergamot, rosemary, thyme, basil, and oregano causes positive sensory notes [10,13,17,18,19]. In addition, the use of citrus fruits or their peels in co-extraction increases the antioxidant activity of olive oil and its nutritional value [18,20,21]. The use of spices in co-malaxation increases the antioxidant activity of olive oils compared to the infusion technique [13]. Tomato by-products are also used in co-milling to enrich olive oils with lycopene [22].

*Thymus mastichina* L. (T. *mastichina*) is an Iberian endemic thyme used in the food industry as a condiment and herbal infusion and as a source of essential oil for the cosmetics industry [23,24]. It is also known as mastic thyme, Spanish marjoram, or white thyme. Traditionally, it was used for treating digestive, respiratory, and rheumatic disorders [25]. The major compounds in the essential oil of *T. mastichina* are 1,8-cineol (58.8–64.1%) and α-terpineol (5.6%) [26]. The main phenolic compounds identified in *T. mastichina* extracts are rosmarinic acid, methoxysalicylic acid, apigenin, kaempferol, luteolin, chlorogenic acid, cafeic acid, and derivatives of luteolin and apigenin [24,27]. Thus, the selection of this species of thyme for co-extraction trials is based on the use of an endogenous plant as well as on its potential, not only in terms of sensory attributes but also for improving the biological value and oxidative stability of enriched flavored oils. Therefore, the aim of this study is to develop a new value-added product, a gourmet flavored oil, based on co-processing overripe healthy ‘Galega Vulgar’ olives, with low intensity of fruity aroma and low amounts of bioactive phenolic compounds, and *Thymus mastichina* L. from organic farming. The effect of the addition of dried thyme, either in olive milling or in malaxation, is investigated. It is expected to obtain a gourmet oil with thyme flavor and, if possible, improved bioactivity and shelf-life. According to our knowledge, this is the first study on co-extraction of olives with this species of thyme and process optimization using response surface methodology.

## 2. Materials and Methods

### 2.1. Biological Material

Portuguese olive fruits of ‘Galega Vulgar’ cultivar used in the present study were produced in a rain-fed olive grove situated in the Beira Baixa region (39°50′ N, 7°42′ W), Portugal. ‘Galega Vulgar’ fruits were picked in January 2021, with a ripening index (RI) of 6.4 and an average weight of 2.5 ± 0.1 g per fruit with a very low water content (42.31% ± 0.21%), were used for the co-processing experiments with thyme addition in the malaxation operation. ‘Galega Vulgar’olives with a RI of 6.2 were used for the co-processing trials with the addition of thyme in the mill. Dried *Thymus mastichina* L. was purchased from Ervas de Zoé, Ladoeiro, Portugal, and was produced according to organic farming (OF) guidelines.

### 2.2. Milling of Thymus Mastichina L. and Particle Size Classification

The dried plants of *T. mastichina* were submitted to milling (hammer mill PX-MFC90D from Kinematic, Switzerland, exit grid of 2 mm opening). Sieve analysis was performed to classify the particles according to their size, using five sieves of Tyler equivalent series (10, 28, 35, 60, and 140 mesh, equivalent to opening sieves of 1.68, 1, 0.42, 0.25, 0.106 mm, respectively)

### 2.3. Co-Processing

Flavored oils were obtained in a laboratory oil mill (Abencor analyzer; MC2 Ingenieria y Sistemas S.L., Seville, Spain), comprising a hammer mill, a malaxation unit, and a cylindrical bowl centrifuge. The olives (c.a. 10 kg) were crushed at 3000 rpm, using a 5 mm grid in the mill.

Co-processing experiments with thyme addition in the malaxator were performed following a central composite rotatable design (CCRD) as a function of the contents of thyme and water [28,29]. In each trial, 0.5 kg of olives was used. In this design, the five levels tested for thyme and water concentration were between 0.4 and 4.6% (*w*/*w*) and between 8.3 and 19.7% (*w*/*w*), respectively (Table 1). Water and thyme were added at the beginning of the malaxation step, carried out at 28−30 °C for 30 min. Paste centrifugation was performed at 3500 rpm for 1 min. After centrifugation, the water traces in the oil were removed with anhydrous sodium sulfate, which was removed by filtration through a cellulose filter (Whatman 41) [30]. After, the oils were collected in amber flasks and stored at 4 °C until analysis.

The combined effects of the concentrations of *T. mastichina* (*Tm*) and of water (*W*) on the oil extraction yield, total phenols, chlorophyll pigments, major fatty acid composition, as well as on chemical quality criteria parameters (acidity, peroxide value, and UV absorbances) of the extracted oils, were investigated by response surface methodology (RSM). RSM allows finding the optimal conditions with a smaller number of experiments than the conventional approach (one variable at a time, OVAT), with the same precision as OVAT and with the advantage of giving information about possible interactions between the variables, which is not possible following the OVAT approach [28,29].

Another set of experiments was carried out with a different batch of olives (RI = 6.2) with the addition of thyme in the mill, at the concentration of the central point of the CCRD (2.5%, *w*/*w*). Water was added to the malaxator at a concentration of 14% (conditions of the central point). The extraction yield was calculated as previously described [31]. Chemical and sensory analyses were performed within one week after extraction.

### 2.4. Shelf-Life Studies

Flavored and unflavoured oil were stored in amber glass bottles at 22−23 °C, in the dark, for 6-month shelf-life studies. These samples were analyzed as described in paragraph 2.6.

### 2.5. Proximate Analysis of Olives and Thyme

Samples (Galega olives after the milling process and thyme plants after milling (2.2)) were subjected to drying at 105 ± 2 °C (Selecta Drying Oven, JP Selecta, Barcelona, Spain) until constant weight, in order to evaluate their moisture content. Fat content was determined by extraction from dried samples in a Soxtec System HT2 Extraction unit (Tecator AB, Hoganas, Sweden), using petroleum ether as an organic solvent. For ash content assay, samples with an initial weight of around 0.5 g were placed in a muffle furnace at 550 °C (Heraeus Instruments, Hanau, Germany). After 24 h, the final weight of the samples was obtained, and the ash content was determined. For protein content assay, nitrogen content (N) was determined in dried samples according to the Kjeldahl method using 0.5 g of each sample (Velp Scientifica UDK 139, Usmate, Italy). The crude protein content was obtained using the conversion factor of 6.25. Mineral element content was determined as follows: approximately 0.4 g of each dried sample were ground in a mortar and weighed in a Teflon tube to which 3 mL of concentrated nitric acid (68%) and 10 mL of concentrated hydrochloric acid (37%) were added. The tubes were then placed in a digestion plate (DigiPrep MS, SCP Science, Quebec, QC, Canada) with a heating cycle to 95 °C that lasted 1 h and remained at 95 °C for 1 h more. After cooling at 20 °C, the samples were filtered into a 25 mL volumetric flask and the volume filled with distilled water. The quantification of the elements (Cu, Zn, Fe, Mn, Na, K, Ca, Mg, P, and S) was done with ICP-OES (Inductively coupled plasma optical emission spectroscopy, Thermo iCAP 7200, Thermo Fisher Scientific, Waltham, MA, USA). Appropriate standards were prepared from a stock solution (100 mg/L) containing the analyzed elements (SCP Science, PlasmaQUAL S22, Baie-D’Urfe, QC, Canada). Results were expressed as mg/kg dry matter (DW).

### 2.6. Chemical and Sensory Characterisation of Flavored Oils

Acidity (% free fatty acids, %FFA, expressed in oleic acid), peroxide value (PV), UV absorbances related with the formation of conjugated hydroperoxides (K_232_) and secondary oxidation products (K_270_), and the major fatty acids (C16:0, C18:0, C18:1, and C18:2) of oils were evaluated by NIR spectroscopy (MPA, Bruker Optics, Ettlingen, Germany). The calibration model B-Olive-Oil (Bruker Optics, Ettlingen, Germany) was used. Spectral information was obtained from olive oil samples, previously prepared obtained at 50 °C (IN600-A, Bruker, Ettlingen, Germany), in absorbance mode and at a wavelength of 1200 to 4000 cm^−^^1^, with 8 cm^−^^1^ resolution and 32 scans.

Total phenols were extracted by liquid-liquid microextraction and evaluated by VIS spectroscopy (JASCO 7800, Jasco Inc., Tokyo, Japan) according to Pizarro et al. [32], as previously described [33]. Results were expressed as milligram of gallic acid equivalent per kilogram of oil (mg GAE/kg). Chlorophyll pigments were assayed in accordance with the IUPAC method proposed by Pokorný et al. [34] using a single beam spectrophotometer (Biochrom Libra S21, Biochrom Ltd., Cambridge, UK) to evaluate the absorbances of oils at 630, 670, and 710 nm against air. The results were expressed as mg pheophytin a/kg oil. All analyses were performed in triplicate.

Samples of flavored oils were also sensory evaluated by a trained panel [35]. A Quantitative Descriptive Analysis (QDA) was applied by using a profile sheet where an unstructured 10 cm length scale was used to mark the intensity of the descriptors [36]. This profile sheet mainly considered the positive attributes (e.g., orthonasal, retronasal, and gustative analysis) of the oils. If defects were found, it was necessary to identify them and quantify their intensity. Oxidative stability (OS) was measured using a Metrohm Rancimat model 670 (Metrohm, Herisau, Switzerland) (temperature of 120 °C; airflow of 20 L h^−1^).

The profile of phenolic compounds was evaluated by high performance liquid chromatography (HPLC) according to the International Olive Council method with some modifications [37]. The phenolic compounds were recovered from the olive oil by liquid-liquid extraction using the procedure proposed by Pirisi et al. [38]. An Agilent 1100 HPLC system (Agilent, Santa Clara, CA, USA), consisting of a degasser, a quaternary pump, a column oven, an autosampler, and a UV detector, was used. The stationary phase was a Purospher C18 analytical column (150 mm × 3.9 mm × 4 µm). The mobile phase consisted of solutions of (A) 0.2% H_3_PO_4_ (*v*/*v*), (B) methanol, and (C) acetonitrile at a constant flow rate of 1 mL min^−1^. The gradient program used was the one indicated by the IOC document [37]. The identification of phenolic compounds was carried out using standards for gallic acid, hydroxytyrosol, tyrosol, vanillic acid, caffeic acid, vanillin, *p*-coumaric acid, *o*-coumaric acid, cinnamic acid, luteolin, apigenin, rosmarinic acid, kaempferol, and pinoresinol. Syringic acid was used as an internal standard. Standards of hydroxytyrosol, tyrosol, vanillic acid, vanillin, caffeic acid, ferulic acid, *o*-coumaric, *p*-coumaric, apigenin, rosmarinic acid were purchased from Sigma-Aldrich (St. Louis, MO, USA), oleuropein and luteolin from Extrasynthese (Genay, France), and pinoresinol from TCI Europe (Zwijndrecht, Belgium).

### 2.7. Statistical Analysis

The obtained results of CCRD, as well as ANOVA (post hoc Tukey test was used; *p* ≤ 0.05), were analyzed using the software Statistica, version 7, from Statsoft, Tulsa, OK, USA. The linear effects, as well as the quadratic effects of each factor (variable) tested (water and thyme concentrations) and of their linear interactions, on each response (extraction yield, total phenols, and chlorophyll pigments contents) were calculated. The significance of each effect was evaluated by analysis of variance. A response surface, described by a first or a second-order polynomial equation, was fitted to each set of experimental results. The first and second-order coefficients of these equations were generated by regression analysis. The goodness of fit of the polynomial models was evaluated by the coefficient of determination (R^2^) and adjusted R^2^ [28,29].

## 3. Results and Discussion

### 3.1. Proximate Analysis of Olives and Thyme

The results for moisture, fat, ash, protein, and mineral elements of olives and *T. mastichina* are presented in Table 2. The low moisture content of the olives highlights the need to perform the optimization of water addition in the malaxation trials, in combination with thyme addition. Galega olives and thyme fat contents (DW) are in accordance to Peres et al. [6] and to Barros et al. [39], respectively. In turn, ash and protein contents of thyme are higher than the values reported by Barros et al. [39]. For mineral composition, both materials have high contents of K (16.04 and 13.82 g/kg, in olives and thyme, respectively) and Ca (3.29 and 11.11 g/kg in olives and thyme, respectively). However, no references on the mineral content in *T. mastichin* a were found. The results obtained in our study are very different from those observed by Kassegn and Mekelle [40] for lemon thyme, where the microelement contents were 0.734, 1.630, 16.41, 0.106 mg kg^−1^ for P, Cu, Fe, and Mn, respectively. However, Kuçukbay and Kuyumcu [41] found values ranging from 6.5−14.90 mg/kg for Cu, 8.470−18.187 g/kg for K, and 8.383−25.570 g/kg for Ca, in other thyme species from Turkey, which are similar to the values obtained in the present study.

### 3.2. Particle-Size Analysis of Milled Thyme

After milling, thyme particles, with dimensions smaller than 2 mm, were separated by fractions using a set of 5 sieves. Figure 1 shows that c.a. 70% of ground material is formed by particles with dimensions between 1 and 1.68 mm, corresponding to the fraction 10/18 mesh. Only 0.2% of the particles have dimensions higher than 1.68 and smaller than 2 mm, and 26.6% have dimensions smaller than 1 mm. Therefore, the milled thyme used in co-processing presented very homogeneous particles concerning their size. These particles are big enough to facilitate their removal from the oil together with the olive pomace by centrifugation in olive oil extraction plants, avoiding emulsion formation and oil loss.

### 3.3. Optimization of Co-Processing Conditions

In the ‘Galega’ cultivar, the oil yield obtained without thyme addition was 10.6%. In CCRD experiments, extraction yields in the presence of thyme varied from 5.1 to 10.7%. Thus, except for experiment 2 (1% thyme and 18% water), where 10.7% oil yield was obtained, the extraction yields obtained in the presence of thyme were lower than in the blank trial.

The statistical analysis of the oil yields showed that the addition of *T. mastichina* had a significant negative linear effect (*p* = 0.014) on oil extraction. It means that an increase in thyme concentration in the malaxation step will promote a decrease in extraction yield. This decrease may be ascribed to oil adsorption to the lignocellulosic material of thyme during malaxation operation, hindering mechanical oil extraction. A similar situation was referred by other authors, which was explained by an absorption/adsorption process [42,43]. For water addition, a positive linear effect was found (*p* = 0.05), meaning that an increase in water concentration will improve the extraction yield. A decrease in the viscosity of olive paste due to the presence of water may help oil extraction since diffusion is promoted. No significant quadratic effect of water and of the interaction effect of thyme and water addition was observed on oil yield. The negative quadratic effect of thyme (*p* > 0.05) was important enough to be retained in the response surface model. Its removal causes a great lack of fit of the model. In experimental design analysis, it is better to retain a “nonsignificant” effect (*p* > 0.05) in the model than to remove an important one [28]. Figure 2 shows the response surface fitted to oil extraction yield, *Y*, as a function of thyme (*Tm*; %, *w*/*w*) and water (*W;* %, *w*/*w*) concentrations. This is a convex surface described by the following second-order polynomial Equation (1):(1)$$Y=6.93+0.335Tm-0.239Tm^{2}+0.221W$$

This model presents a good fit to the experimental points since it has a determination coefficient, R^2^, of 0.69 and an adjusted determination coefficient (R^2^_Adj_) of 0.59. Therefore, 69% of the experimental results are explained by this model.

Higher oil yields were observed for higher water contents and lower thyme concentrations (Figure 2).

### 3.4. Characterization of Flavored Oils

The results of chemical quality criteria (acidity, PV, and UV absorbances) and major fatty acid composition (C16:0; C18:1; C18:2, and C18:3) of (i) the flavored oils obtained from CCDR trials, (ii) of one flavored oil obtained by co-processing with thyme addition during milling, and (iii) of the virgin olive oil obtained under the same extraction conditions without thyme addition, are presented in Table 3. The variation observed in chemical quality criteria of flavored oils, acidity (0.17−0.21%), PV (4.2−4.9 meq O_2_ kg^−1^), K_232_ (1.54−1.63), and K_270_ (0.11−0.15), is not significant. This shows that the co-processing with *T. mastichina* did not affect the quality of the flavored oils obtained. Moreover, a similar behavior was observed concerning the major fatty acid contents: palmitic acid varied from 11.54 to 12.22%; oleic acid ranged from 77.00 to 77.42%; linoleic acid varied from 4.56 to 5.06% and linolenic acid from 0.5 to 0.8%. In addition, flavored oils obtained by co-processing, either with thyme addition in the hammer mill or in the malaxator, have similar quality values and fatty acid composition. Other authors have stated that when the initial indices are relatively high, the obtained flavored oils may present quality parameters values above the legal limit for EVOO [17,44]. This was not observed in our study.

The amounts of total phenolic compounds (*TPH*) in flavored oils varied from 60.7 to 141.6 mg GAE/kg oil, while the VOO obtained without co-processing with thyme had a *TPH* of 71.4 ± 6.8 mg GAE/kg oil. In fact, not all the flavored oils obtained by co-processing with *T. mastichina* added in the malaxator presented higher amounts of phenolic compounds than the original VOO. The low content of *TPH* in VOO is explained by the high ripening stage of the olives used, together with a relatively low *TPH* of Galega VOO, when compared to VOO from other cultivars at the same RI [6,45]. The flavored oil obtained under the conditions of the central point (Table 1: 2.5% thyme and 14% water) showed a *TPH* of 115.4 ± 4.9 mg GAE/kg oil, corresponding to a *TPH* increase of 61.6% when compared with the non-flavored VOO. The co-processed flavored oil obtained by the addition of thyme in the mill, under the conditions of the central point, presented a *TPH* of 189.66 ± 6.8 mg GAE/kg oil, which represents an increase of 75% with respect to the original VOO. Thus, phenolic compounds extraction seems to be more efficient by co-processing with *T. mastichina* added in the milling than in the malaxation operation. These results show the importance of performing optimization trials before the addition of the flavoring agent.

Data analysis of CCRD showed that the content of phenolic compounds in flavored oils linearly increased with thyme concentration (*p* = 0.004) and with water added in the malaxator (*p* = 0.06). A significant negative quadratic effect of water concentration (*p* = 0.045), indicating a convex quadratic response as a function of this effect, was also found. No significant effects of thyme at quadratic level or of the interaction thyme x water were found. Therefore, a convex response surface, described by the following second-order polynomial, can be fitted to *TPH* (mg GAE/kg) as a function thyme (*Tm*; %, *w*/*w*) and water (*W;* %, *w*/*w*) concentrations (Figure 3a), Equation (2):(2)$$TPH=-98.19+12.91Tm+22.99W-0.732W^{2}$$

The good fit of the model is demonstrated by high values of both R^2^ (0.78) and R^2^_Adj_ (0.71).

Concerning chlorophyll pigments, their content in flavored oils with *T. mastichina* added in malaxation varied from 55.8 to 67.9 mg pheophytin a/kg oil against 49.4 mg pheophytin a/kg oil in Galega VOO obtained from the same fruits. These results indicate that the migration of green pigments from the thyme to the oils occurs during co-extraction. In fact, CCRD data analysis showed that the content of chlorophyll pigments in the oils significantly depends on the thyme amount (*p* = 10^−6^) added during malaxation, increasing with it. The water added in the malaxator showed to have a negative linear effect (*p* = 0.06) and a negative quadratic effect (*p* = 0.008) on green pigment extraction to the oils. Thus, chlorophyll pigments in flavored oils can be described by a convex surface as a function of thyme and water concentrations in pastes during malaxation (Figure 3b). This response surface is given by the following second-order polynomial equation where the amount of chlorophyll pigments (*CP*; mg/kg oil) is a function of thyme content (*Tm*; %, *w*/*w*) and water (*W;* %, *w*/*w*) concentrations, Equation (3):(3)$$CP=39.65+2.70Tm+2.39W-0.092W^{2}$$

This model has a very good fit to the experimental results (R^2^ = 0.96; R^2^_Adj_ = 0.94).

Figure 3 shows that both *TPH* and chlorophyll pigments are described by similar shape convex response surfaces. The highest *TPH* values are obtained for co-processing with thyme concentrations higher than c.a. 3.5−4.0% and water contents in the range c.a. 14−18%. The highest pigment contents are also obtained with thyme concentrations higher than c.a. 3.5% and water content of the pastes between 10 and 18%. Conversely, the highest extraction yields are observed for thyme contents lower than 2.5% and when high water contents are used (>18%) (Figure 2).

According to several authors, the addition of thyme (T. *vulgaris*) in olive oil increases the *TPH* [46,47], especially when the co-extraction technique is used [19]. The presence of high amounts of water in olive pastes during malaxation has been related to the loss of hydrophilic phenols in the water phase [1,48]. Therefore, in our study, the production of high-quality flavored oils enriched with bioactive phenolic compounds extracted from *T. mastichina* was obtained under conditions that will conduct to lower oil extraction yields. The use of RSM showed to be a useful tool to find the best operation conditions for flavored oil production by co-extraction. The evaluation of the profiles of oil yield, *TPH*, and chlorophylls in oils obtained under different co-extraction conditions (water and thyme concentrations) was only possible via the visual observation of each response surface fitted to each data set. The optimal co-extraction conditions were chosen from the information shown in these response surfaces.

### 3.5. Shelf-Life Studies: Quality, Phenol Composition, Sensory Analysis, and Oxidative Stability

The VOO and flavored oil samples obtained by co-processing with *T. mastichina*, added either in the malaxation or in milling operations, under the conditions of the central point of the CCRD (Table 1), were submitted to shelf-life studies. After 6 months of storage in the dark at 22−23 °C, VOO and flavored oils were assayed for their chemical parameters and sensory properties. No significant differences were observed among VOO and flavored oils in terms of quality parameters and fatty acid composition (Table 4). Moreover, for each quality parameter, no significant differences were observed between the initial values and those obtained after 6 months of storage, except for PV, which showed around a 66% increase in stored oils. This indicates that, after 6 months of storage in the dark, oil oxidation was at the initial induction stage of hydroperoxide formation. Along this storage period, *TPH* and chlorophyll pigments decreased 8.6 and 7.6%, respectively.

In order to identify the phenols that migrated from the T. *mastichina* to the oils, increasing their bioactivity, an HPLC profile at 280 nm (all phenol compounds identified by standards), 320 nm (*p*-coumaric acid), and 360 nm (flavonoids) was performed. The chromatographic phenolic profiles at 280 nm of thyme-flavored oils, compared to unflavored VOO, are shown in Figure 4.

In our study, as a result of the co-extraction with *T. mastichina*, none of the VOO phenolic compounds disappeared in flavored oils. An increase in some phenolic compounds, already present in VOO, as well as the presence of new compounds, were observed. In fact, flavored oils showed an increase in vanillin (peak 5), *p*-coumaric acid (peak 6), luteolin (peak 14), apigenin (peak 15), an unidentified peak at a retention time (RT) at 40 min (peak b: non-flavonoid, not detected at 360 nm), and an unidentified peak at RT 50 min (peak c) already present in VOO.

As reported by other studies, the main phenolic compounds identified in *T. mastichina* extracts are rosmarinic acid, methoxysalicylic acid, apigenin, kaempferol, luteolin, chlorogenic acid, cafeic acid, and derivatives of luteolin and apigenin [24,27]. The HPLC profiles of standards show that rosmarinic acid has an RT of 31.67 min and higher absorbance at 320 nm. In flavored samples of our study, rosmarinic acid showed not to be the main phenolic compound transferred from the thyme to the oil (Figure 4). Chlorogenic acid (RT = 16.97 min), caffeic acid (RT = 18.8 min), and kaempferol (RT = 43.13 min), referred to in the literature as present in *Thymus mastichina*, were not transferred to the oil.

Peaks 10 and 16 are referred in the literature as 3,4-DHPEA-EDA (oleacein or dialdehydic form of elenolic acid linked to hydroxytyrosol) and *p*-HPEA-EA (ligstroside aglycone, aldehyde, and hydroxylic form), respectively [37]. Comparing the profile of unflavored Galega VOO obtained in our study with those from the literature [6], higher amounts of 3,4-DHPEA-EDA and *p*-HPEA-EA were observed in VOO from fruits with low ripening indices, which are characterized by high phenol content and flavor intensity.

The sensory analysis was also performed for the original VOO and flavored oils obtained under the conditions of the central point of the CCRD, either by the addition of *T. mastichina* in the mill or in the malaxator. These oils were sensory evaluated immediately after extraction and after 6 months of storage (Figure 5). After extraction, Galega VOO exhibited a very low (<2) intensity of ripe fruity (orthonasal and retronasal evaluation), bitterness, and pungency. Moreover, defects ascribed to frostbitten olives (wet wood) were detected by assessors with an intensity lower than 3.5. Therefore, despite chemical quality criteria corresponding to the extra virgin olive oil category (Table 4), due to the identification of this defect, the oil can no longer be classified as EVOO but falls in the category of “virgin olive oil”. The presence of this sensory defect has already been reported in VOO from fruits after weather-related hazards like frost [49,50]. After 6 months of storage in the dark at 22−23 °C, the defect was perceived with similar intensity; a decrease in the intensity of ripe fruity (orthonasal and retronasal) was registered in VOO, while bitter and pungent attributes were no longer detected.

In the flavored oils with *Thymus mastichina*, the defect of “wet wood” perceived in VOO was not detected. A thyme flavor (orthonasal and rethonasal), with very high intensity (≥8) for the hammer mill co-processed oils, and of intensity 5−6, in malaxator co-processed flavored oils, was registered. Moreover, the olive fruity (orthonasal and rethonasal) notes, characteristic of EVOO, were almost absent in thyme-flavored oils. These results indicate that strong changes in the volatile fraction occurred due to the co-processing of olives with thyme, as previously reported by others [51].

After 6 months of storage, no defects were detected in flavored oils. However, a decrease in thyme flavor was observed in both flavored oils, being more pronounced in the oil flavored by the addition of thyme in milling. The bitter taste had similar intensity in both fresh flavored oils. After 6 months of storage, the bitter taste was absent in the flavored oil with thyme addition during malaxation, while its initial intensity was maintained in the flavored oil with thyme addition in milling. Concerning the initial intensity of pungency, it was higher in the flavored oil obtained by thyme addition in milling, but, at the end of the storage experiment, the intensity of this attribute was similar in both flavored oils.

In conclusion, the flavored oil obtained by co-processing with *T. mastichina* in the mill showed higher initial intensity scores for positive attributes but, after 6 months of storage, its thyme flavor was less intense than that of the oil obtained by co-processing with thyme addition in the malaxator.

In Figure 6, the results of oxidative stability (OS) of VOO and flavored oils by co-extraction with *T. mastichina* added either in the mill or in the malaxator are presented. Co-processing with the addition of thyme in a hammer mill (FTh) improves the OS of the flavored oils (14.07 h in VOO vs. 19.03 h in flavored oil), while no significant differences were found between VOO and thyme flavored oils in the malaxation step (FT) (14.30 h). After six months of storage, the oils showed a significant decrease in OS. No differences were observed between the VOO and the flavored oil with the addition of thyme in malaxation (c.a. 10.3 h), but the OS of the flavored oil obtained by co-extraction with *T. mastichina* added in the mill was higher, with a value of 16.06 h (corresponding to a 15.6% reduction after 6 months of storage).

In Rubió et al. [52], the addition of *Thymus zygis* extracts significantly improved the oxidative stability of oils when compared to VOO, while Issaoui et al. [47] only detected a nonsignificant increase in OS of flavored oils. Conversely, Issaoui et al. [53] found that the oxidative stability of oils decreased with the addition of thyme, regardless of the concentration added (20 to 80 g/kg).

Depending on several factors, such as the type, the level, and the method of incorporation of the flavoring agent, the initial properties of the olive oil, as well as the storage conditions, results in increasing or decreasing trends in OS have been reported [10,16,54,55,56]. Our results confirm that direct malaxation of the olive paste with thyme or its addition in the hammer mill is a technique that is easy to carry out. Moreover, it does not include any additional unit operation to the conventional olive oil extraction process and is faster than infusion. In addition, both flavoring techniques are conventional green processes that do not require the use of any organic solvents [13].

## 4. Conclusions

Consumer’s attitude towards olive oil is changing. Their search for different flavored oils that match several food pairings is an opportunity for the olive oil sector. Thus, the research on developing adequate technologies to produce these novel oils, presenting adequate sensory properties, improved biological value, and shelf-life stability, is of utmost importance for the olive oil industry. Although different techniques can be used, some of them consisting of the incorporation of the flavoring agent in the olive oil may have a deleterious effect on nutritional and shelf-life characteristics. This study showed the feasibility of sustainably producing new high-value products obtained with oils from autochthonous olive cultivars using natural resources. Olive oils obtained from overripe olives, presenting a slight aroma and low amounts of bioactive phenolic compounds, can be highly valorized by co-extraction with thyme. The optimization of co-extraction and storage studies is a good way to evaluate the main effects of the addition of thyme. The co-processing of ripe olives with *T. mastichina* added, either in the milling or in the malaxation operations, showed to be a feasible and easy technique to obtain flavored oils with intense thyme sensory notes and without sensory defects that were present in the VOO extracted from the same olives. Considering chemical quality criteria, these flavored oils have the characteristics of VOO, but they cannot have this classification due to legislation issues. Under optimized co-processing conditions, flavored oils presented higher phenolic contents, and higher biologic value, than the non-flavored VOO. The flavored oil obtained by co-processing with *T. mastichina* addition in the mill showed higher oxidative stability than the VOO and the co-processed oil with thyme addition in the malaxator, even after six-month storage in the dark.

## Figures and Tables

**Figure 1 life-11-01048-f001:**
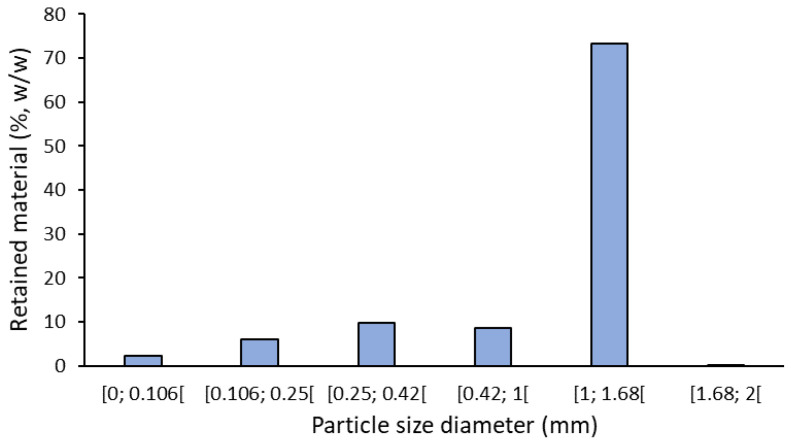
Histogram of different granulometric fractions of milled thyme.

**Figure 2 life-11-01048-f002:**
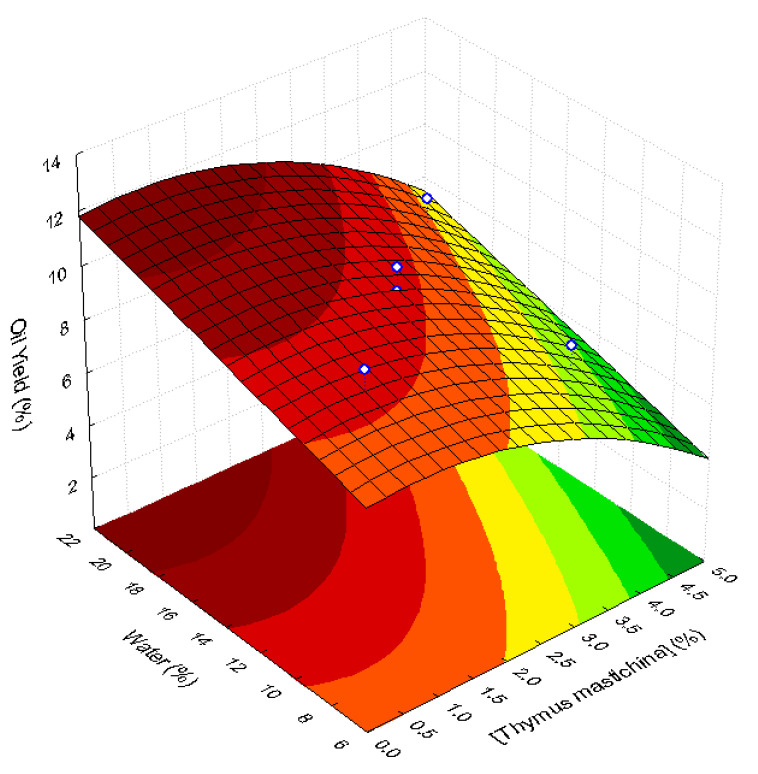
Response surface describing oil yield (%) as a function of water (%, *w*/*w*) and thyme concentration (%, *w*/*w*), added to ‘Galega Vulgar’ fruits during malaxation.

**Figure 3 life-11-01048-f003:**
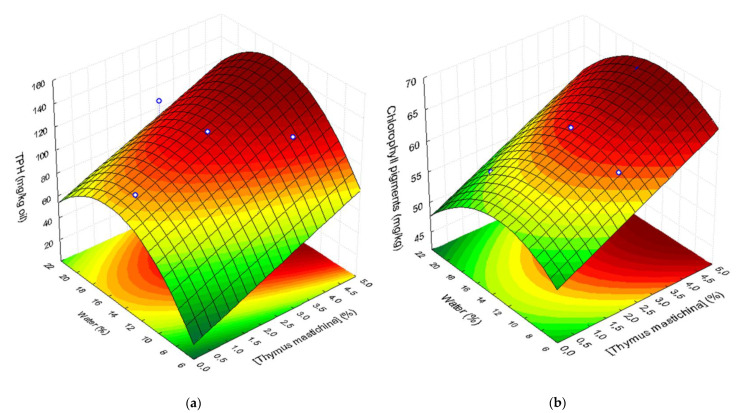
Response surfaces describing (**a**) total phenols (TPH; mg GAE/kg), and (**b**) chlorophyll pigments in flavored oils as a function of water (%, *w*/*w*) and thyme concentration (%, *w*/*w*), added to ‘Galega Vulgar’ fruits during malaxation.

**Figure 4 life-11-01048-f004:**
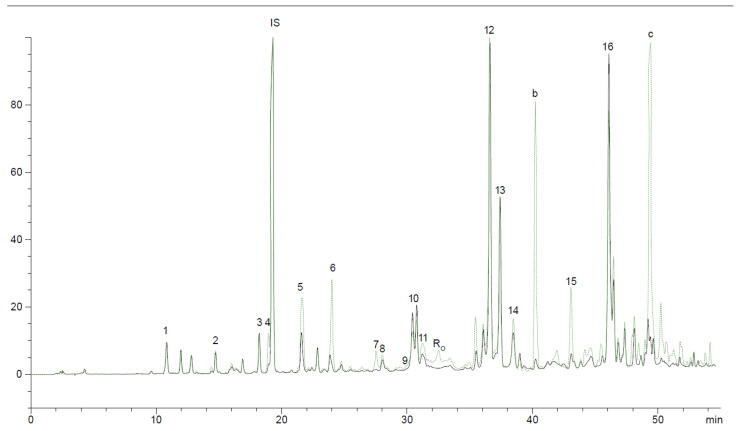
HPLC-RP-VWD 280 nm phenolic profile of the unflavored VOO (continuous line) and flavored oil (dotted line). (1—hydroxytyrosol; 2—tyrosol; 3—vanilic acid; IS—Internal standard (syringic acid) 4—caffeic acid; 5—vanillin; 6—*p*-coumaric acid; 7—unknown; 8—luteolin-7-glucosid; 9—*o*-coumaric acid; 10—3,4-DHPEA-EDA (literature); 11—oleuropein; Ro- Rosmarinic acid; 12—pinoresinol; 13—cinnamic acid 14—luteolin; 15—apigenin; 16—*p*-HPEA-EA (literature); b and c—unknown).

**Figure 5 life-11-01048-f005:**
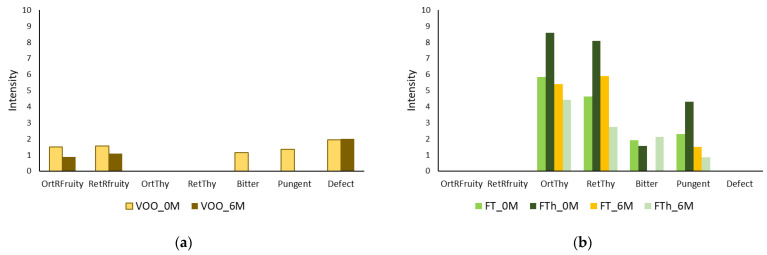
Sensory evaluation of (**a**) the VOO without storage (VOO_0M) and after 6 months storage at 22−23 °C in the dark (VOO_6 M), and of (**b**) thyme flavored oils in the malaxation step (FT_0M) and in the hammer mill (FTh_0M) without storage and after 6 months storage at 22−23 °C in the dark (FT_6 M and FTh_6M, respectively). OrtRFruity—orthonasal ripe olive fruity; RetRfruity—retronasal ripe fruity; OrtThy—orthonasal thyme; RetThy—retronasal thyme.

**Figure 6 life-11-01048-f006:**
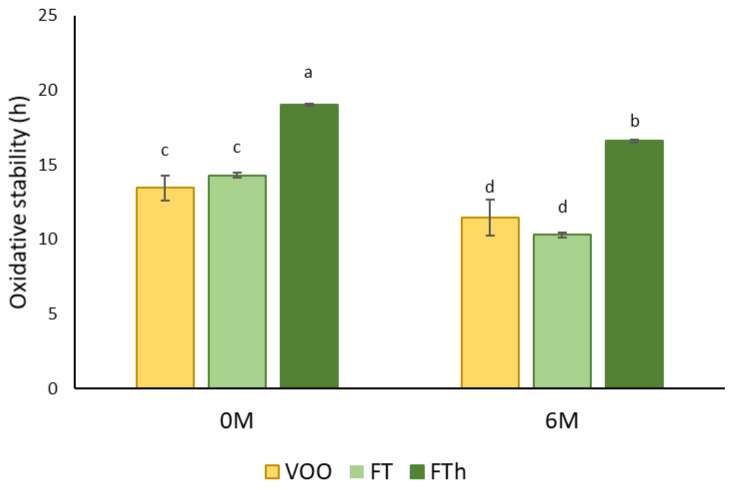
Oxidative stability (hour) of VOO and thyme flavored oils obtained by co-extraction in the malaxation (FT) or in the mill (FTh) at time 0 (0M) and after 6 months of storage (6M) at 22−23 °C in the dark (different letters of each bar (a, b, c, d) indicate significant differences in the oxidative stability of the oils at *p* < 0.05 according to Tukey test).

**Table 1 life-11-01048-t001:** CCRD followed in the experiments for co-processing as a function of the amounts of thyme and water added in malaxation. Experiment Nº 14 (control) corresponds to the virgin olive oil obtained under the same extraction conditions as flavored oils.

Experimental Points	Experiment Number	[Thyme] Coded Value	[Water] Coded Value	[Thyme] (%) Decoded Values	[Water] (%) Decoded Values
Factorial points	1	−1	−1	1	10
	2	−1	1	1	18
	3	1	−1	4	10
	4	1	1	4	18
Star points	5	$-\sqrt{2}$	0	0.4	14
	6	$\sqrt{2}$	0	4.6	14
	7	0	$-\sqrt{2}$	2.5	8.3
	8	0	$\sqrt{2}$	2.5	19.7
Central points	9	0	0	2.5	14
	10	0	0	2.5	14
	11	0	0	2.5	14
	12	0	0	2.5	14
	13	0	0	2.5	14
Control	14	_	_	-	14

**Table 2 life-11-01048-t002:** Proximate analysis (DW) of olives and *Thymus mastichina* L.

Parameter	Unit	Olives	*T. mastichina*
Moisture	(%)	42.31± 0.21	6.30 ± 0.04
Fat	(%)	39.62± 0.82	3.08 ± 0.01
Ash	(%)	3.67 ± 0.59	5.35 ± 0.06
Protein	(%)	5.00 ± 0.62	7.79 ± 0.15
Cu		9.2 ± 1.3	10.1 ± 1.1
Zn		19.8 ± 3.0	83.2 ± 18.5
Fe		91.0 ± 6.4	218.5 ± 4.5
Mn		23.8 ± 1.3	292.7 ± 44.5
Na		154.8 ± 7.6	205.5 ± 4.8
K	(mg/kg)	16040 ± 199	13822 ± 1195
Ca		3288 ± 101	11118 ± 464
Mg		472 ± 7	2178 ± 214
P		1199 ± 17	1958 ± 260
S		853 ± 33	1854 ± 107

**Table 3 life-11-01048-t003:** Results of acidity (% oleic acid), peroxide value (meq O_2_ kg^−1^), UV absorbances (K_232_, K_270_), palmitic acid (%) (C16:0), oleic acid (%) (C18:1), linoleic acid (%) (C18:2), and linolenic acid (%) (C18:3) (conditions of each experiment are shown in Table 1; experiment 14 corresponds to the co-processed flavored oil obtained by adding thyme in the hammer mill; experiment 15 is the VOO extracted from the same fruits without thyme addition, i.e., the control).

Experiment	Acidity	PV	K_232_	K_270_	C16:0	C18:1	C18:2	C18:3
1	0.17	4.5	1.54	0.13	11.59	77.03	5.06	0.48
2	0.18	4.3	1.57	0.11	11.54	76.99	5.00	0.58
3	0.19	4.5	1.60	0.13	12.04	77.31	4.73	0.76
4	0.21	4.3	1.59	0.12	12.06	77.21	4.66	0.75
5	0.18	4.2	1.58	0.11	11.59	77.09	4.86	0.59
6	0.18	4.7	1.61	0.15	12.22	77.42	4.56	0.82
7	0.21	4.7	1.62	0.14	11.85	77.07	4.85	0.67
8	0.17	4.3	1.60	0.12	11.97	77.10	4.63	0.67
9	0.18	4.5	1.61	0.14	11.74	77.28	4.57	0.70
10	0.17	4.9	1.61	0.14	11.87	77.16	4.73	0.72
11	0.19	4.2	1.61	0.14	11.93	77.04	4.73	0.70
12	0.18	4.7	1.63	0.14	11.97	77.08	4.77	0.71
13	0.18	4.6	1.61	0.14	11.88	77.14	4.70	0.73
14 (hammer mill)	0.11	4.5	1.72	0.18	11.93	77.02	4.78	0.81
15 (VOO-control)	0.20	4.4	1.59	0.12	11.47	76.94	4.86	0.62

**Table 4 life-11-01048-t004:** Results of acidity (% oleic acid), peroxide value (meq O_2_kg^−1^), UV absorbances (K_232_, K_270_), palmitic acid (%) (C16:0), oleic acid (%) (C18:1) and linoleic acid (%) (C18:2) of the co-processed flavored oils obtained by adding thyme in the malaxator (experiments 9–13. corresponding to the central point of Table 1) or in the hammer mill (experiment 14); and of the virgin olive oil extracted from the same fruits without thyme addition (experiment 15), after 6 months storage at 22−23 °C in the dark.

Experiment	Acidity	PV	K_232_	K_270_	C16:0	C18:1	C18:2	C18:3	TPH
9	0.21	7.39	1.74	0.18	11.90	2.60	4.78	0.61	120.08
10	0.20	7.07	1.74	0.16	11.84	2.58	4.92	0.58	108.19
11	0.23	6.89	1.76	0.16	11.88	2.60	4.94	0.60	97.33
12	0.22	7.65	1.78	0.17	11.95	2.63	4.96	0.63	107.13
13	0.22	7.58	1.77	0.16	11.83	2.57	4.97	0.9	99.28
14 (hammer mill)	0.25	6.7	1.78	0.19	11.83	2.62	4.62	0.70	158.60
15 (VOO-control)	0.25	7.5	1.78	0.15	11.59	2.45	5.14	0.50	69.42

## Data Availability

Data are available upon request to the authors.

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
