# Peer review of "Co-Processed Olive Oils with Thymus mastichina L.—New Product Optimization"

_life, 2021, doi:10.3390/life11101048_

Round 1

Reviewer 1 Report

No comment

Author Response

Dear Colleague,

We would like to thank you for taking the time to review our manuscript submitted to Life and for your positive evaluation.

Best regards,

Suzana Ferreira-Dias

Reviewer 2 Report

In the present paper, the authors have amply demonstrated that the co-processing of ripe olives with T. mastichina, both in the milling and in the malaxation operations, is a practicable and easy technique for obtaining gourmet oils with intense thyme sensory notes and without sensory defects. These oils have been characterized on the basis of chemical quality criteria, for phenolic content and biological value. In addition, shelf life tests were carried out highlighting that the flavored oil obtained from the co-processing with the addition of thyme in the mill has a greater stability to oxidation than the VOO and the co-processed oil with thyme addition in the malaxator, even after six months of storage in the dark. Although the aromatization of oils with spices  and how it increases the content of total phenols has already been described, on the whole, the paper is well organized, clearly written, and arouses interest to the readers.

Author Response

Dear Colleague,

 We would like to thank you for taking the time to review our manuscript submitted to Life and for your positive comments.

Best regards,

Suzana Ferreira-Dias

Reviewer 3 Report

The manuscript of Peres and co-workers deals with olive oils with Thymus mastichina in an attempt to develop a new value-added product, a gourmet flavored oil, based on co-processing process. Specifically, the manuscript regards the interesting topic related to fabrication, characterization of new food products supplemented by bioactive compounds and in particular by the fortification of olive oil.

The paper appears interesting and well written. However, there are some details and weakness that should be analyzed. The introduction section does not provide the properly description of olive oil and the field of olive oil enrichment. An important issue related to the co-procession method and the food grade products proposed should take in consideration e described by means of well supported literature. Moreover, the authors could better focus the relevance of this work to new findings in this field (fortification, enrichment, addition of bio compounds…), and the ways in which the new results have advanced the specific field of olive oil treatment. As a whole I suggest a revision and further reconsideration (see below for details).

Introduction: more details could be provided to help the readers (also taking in consideration the readership of the specific journal Life) to better focus on the detailed fields: olive oil and fortifications via bioactive compounds. I would suggest to implement this part also by referring to recent specific references concerning the description of olive oil and enrichment of olive oils via polyphenols and other bioactive compounds and so on. See: doi: 10.3390/colloids4030038; 10.1016/j.freeradbiomed.2007.11.010; 10.1371/journal.pone.0176580; 10.3390/colloids3030059 and so on.

Methods. Page 2 lines124-127 “After centrifugation, the water traces in the oils were removed with anhydrous sodium sulfate which was removed by filtration through a cellulose filter... After, the oils were collected in amber flasks and stored at 4°C 126 until analysis.” This is an important point, (!) are the author sure that the addition of anhydrous sodium sulfate could be used for a food grade product? Additionally, the filtration represents a further certainly not time and material sparing. I suggest to well highlight this point in the text, especially when in the whole text was claimed that this new product could be safely used in the food industry.

Results: page 7. Could you please provide further details concerning the size distribution?

Results: page 7-9, the significance of these data, the connection with this new finding as well the connection with other systems is not clear to me. Please try to better connect these data with the whole message of this study.

Conclusions: some statements reported in this section are the same reported in the abstract section. Please try to differentiate and avoid to report the same information twice. In this section the authors should try to emphasize the relevance of this work to new findings in this field and the ways in which the new results have advanced the field.

Author Response

Dear Colleague,

We would like to thank you for taking the time to review our manuscript submitted to Life and for your comments that helped us to improve its quality.

For your convenience, we have included below the itemized list or your comments and our replies according to changes and modifications that have now been made to the original manuscript (highlighted in yellow).

We hope that you will find our manuscript improved, and that you now feel it has the required quality for publication in the Special Issue of Life “Advances in Edible Oil Processing”.

Best regards,

Suzana Ferreira-Dias

Comments and Suggestions for Authors

Comment 1: The manuscript of Peres and co-workers deals with olive oils with Thymus mastichina in an attempt to develop a new value-added product, a gourmet flavored oil, based on co-processing process. Specifically, the manuscript regards the interesting topic related to fabrication, characterization of new food products supplemented by bioactive compounds and in particular by the fortification of olive oil. The paper appears interesting and well written. However, there are some details and weakness that should be analyzed. The introduction section does not provide the properly description of olive oil and the field of olive oil enrichment.

Introduction: more details could be provided to help the readers (also taking in consideration the readership of the specific journal Life) to better focus on the detailed fields: olive oil and fortifications via bioactive compounds. I would suggest to implement this part also by referring to recent specific references concerning the description of olive oil and enrichment of olive oils via polyphenols and other bioactive compounds and so on. See: doi: 10.3390/colloids4030038; 10.1016/j.freeradbiomed.2007.11.010; 10.1371/journal.pone.0176580; 10.3390/colloids3030059 and so on.

Ans: Thank you very much for your comments. All the modifications are highlighted in yellow in the manuscript.

This manuscript is to be published in the Special Issue of Life, “Advances in Edible Oil Processing”, which is focused on oil processing field. Therefore, it is expected that all the readers know what virgin olive oil is. However, as you suggested, we added the following sentence at the beginning of the Introduction:

“Virgin olive oil is the oil extracted from the fruits of the olive tree (Olea europaea L.) using exclusively mechanical extraction techniques under conditions that will not affect the original composition of the oil.”

Concerning the “field of olive oil enrichment via polyphenols and other bioactive compounds”, the introduction contains 13 references (10 to 22) about co-extraction (co-processing) and olive oil enrichment with herbs, vegetables, spices and extracts (lines 48-79). These references were considered by us as the most important to support our study. This manuscript is not a review in the field of olive oil enrichment but is focused only on co-processing of olives. This paper has a reasonable number of references (56) and we believe that more references could be not adequate for a research article.

The field of olive oil is so wide that, in Google scholar, from 2015-2021, about 123 000 publications appear, while for “Olive oil fortification & bioactive compounds”, there are around 5990 publications for the same period.

The references you suggested are the following:

  • Atkinson et al., Tocopherols and tocotrienols in membranes: A critical review, Free Radical Biology & Medicine 44 (2008) 739–764
  • Cinelli et al. (2019), Red Wine-Enriched Olive Oil Emulsions: Role of Wine Polyphenols in the Oxidative Stability, Colloids Interfaces 2019, 3(3), 59; https://doi.org/10.3390/colloids3030059
  • D’Amato et al. Biofortification (Se): Does it increase the content of phenolic compounds in virgin olive oil (VOO)? PLoS ONE (2017) 12 (4): e0176580. https://doi.org/10.1371/journal. pone.0176580
  • Cinelli et al., Veiled Extra Virgin Olive Oils: Role of Emulsion, Water and Antioxidants, Colloids Interfaces (2020), 4, 0038; doi:10.3390/colloids4030038

Concerning these references, we apologize but they do not seem adequate for the introduction of our manuscript, and we decided not to include them due to the following reasons:

  • Atkinson et al (2008) concerns tocopherols and tocotrienols in membranes and not in olive oil. Moreover, we did not evaluate the content of these compounds in our samples.
  • Cinelli et al. (2019) concerns Olive Oil Emulsions, while we are producing an oil and not an emulsion.
  • D’Amato et al. (2017) concerns olive orchard fertilization with selenium and the effects on the phenolic profile of Extra Virgin Olive Oil (EVOO).
  • Cinelli et al. (2020) is a literature review about Veiled Extra Virgin Olive Oils. It is a very interesting field we would like to consider in further research. However, this review is focused on O/W emulsions using veiled EVOO. In our case, we do have neither emulsion nor veiled EVOO because our oils were filtered before bottling and analysis.
  •  
  • Comment 2: An important issue related to the co-procession method and the food grade products proposed should take in consideration described by means of well supported literature. Moreover, the authors could better focus the relevance of this work to new findings in this field (fortification, enrichment, addition of bio compounds…), and the ways in which the new results have advanced the specific field of olive oil treatment. As a whole I suggest a revision and further reconsideration (see below for details).

Ans: Thank you for your suggestions.

The following information was added to Introduction section (lines 62-63):

Vegetables, aromatic herbs, fruits, nuts, essential oils, and spices are the most common ingredients used, added either as infusion, ethanolic extracts of essential oils or by co-processing [10-16].

In fact, we would like to clarify that the aim of this study was to produce a novel product using overripe healthy olives, which are characterized by low olive flavour notes and low phenolic content, using co-extraction with a thyme from the Iberian region, to obtain a gourmet oil with thyme flavour and, if possible, improved bioactivity and shelf-life. The approach of this study is technological, with process optimization, and not focused on medical and pharmaceutical field.

The aim of the study was improved as follows (lines 91-98):

“Therefore, the aim of this study is to develop a new value-added product, a gourmet flavored oil, based on co-processing overripe healthy ‘Galega Vulgar’ olives, with low intensity of fruity aroma and low amounts of bioactive phenolic compounds, and Thymus mastichina L. from organic farming. The effect of the addition of dried thyme, either in olive milling or in malaxation, is investigated. It is expected to obtain a gourmet oil with thyme flavour and, if possible, improved bioactivity and shelf-life. According to our knowledge, this is the first study on co-extraction of olives with this species of thyme and process optimization using response surface methodology.”

Comment 3: Page 2 lines124-127 “After centrifugation, the water traces in the oils were removed with anhydrous sodium sulfate which was removed by filtration through a cellulose filter... After, the oils were collected in amber flasks and stored at 4°C 126 until analysis.” This is an important point, (!) are the author sure that the addition of anhydrous sodium sulfate could be used for a food grade product? Additionally, the filtration represents a further certainly not time and material sparing. I suggest to well highlight this point in the text, especially when in the whole text was claimed that this new product could be safely used in the food industry.

Ans: The addition of anhydrous sodium sulphate is to remove water only for analytical purposes, as described in the standard ISO 661:2003 (Animal and vegetable fats and oils - Preparation of the test sample"). This reference ([30]) was added to the manuscript:

Lines 129-131: “After centrifugation, the water traces in the oils were removed with anhydrous sodium sulfate which was removed by filtration through a cellulose filter (Whatman 41)[30].

Again, in our study, filtration was used prior to analysis. In addition, olive oil filtration is a current practice in olive oil extraction plants to avoid the formation of sensory defects during olive oil storage.

Comment 4: Results: page 7. Could you please provide further details concerning the size distribution?

Ans: We do not understand what you mean by “further details concerning the size distribution”. This characterization showed that thyme particles are very homogeneous concerning their size: 70 % of ground material is formed by particles with dimensions between 1 and 1.68 mm. This is important because small particles would be difficult to be removed from the oil and could also promote emulsions with subsequent oil yield decrease. This information was added to the text, as follows (lines 266-269):

“These particles are big enough to facilitate their removal from the oil together with the olive pomace by centrifugation, in olive oil extraction plants, avoiding emulsion formation and oil loss.”

Comment 5: Results: page 7-9, the significance of these data, the connection with this new finding as well the connection with other systems is not clear to me. Please try to better connect these data with the whole message of this study.

Ans: We do not understand your concerns. Results from page 7 to 9 refer product optimization following a central composite rotatable design (CCRD) as a function of thyme and water amounts added during malaxation co-processing. In Food Technology, response surface methodology (RSM) has been widely used for process and product optimization. These methodologies allow finding the optimal conditions with a smaller number of experiments than the conventional approach (one variable at a time, OVAT), with the same precision as OVAT and with the advantage of giving information about possible interactions between the variables which not possible in OVAT. As far as we know, this is the first study on optimization of co-processing of olives using RSM. The background of RSM is explained in references 28 and 29:

  1. Haaland, P.D. Experimental Design in Biotechnology, ; Marcel Dekker Inc.: New York, NY, USA 1989.
  2. Montgomery, D.C. Design and Analysis of Experiments, 3rd ed.; John Wiley & Sons: New York, 2017.

The following information was added to 2.3. Co-processing section (lines 139-142):

“RSM allows finding the optimal conditions with a smaller number of experiments than the conventional approach (one variable at a time, OVAT), with the same precision as OVAT and with the advantage of giving information about possible interactions between the variables which not possible following OVAT approach [28, 29].”

In an attempt to connect experimental data with the conclusions of our study, the following sentences were re-written as follows (lines 383-394):

“According to several authors, the addition of thyme (T. vulgaris) in olive oil increases the TPH [46,47], especially when the co-extraction technique is used [19]. The presence of high amounts of water in olive pastes during malaxation has been related with the loss of hydrophilic phenols in the water phase [1,48]. Therefore, in our study, the production of high-quality flavored oils enriched with bioactive phenolic compounds extracted from T. mastichina was obtained under conditions that will conduct to lower oil extraction yields. The use of RSM showed to be a useful tool to find the best operation conditions for flavored oil production by co-extraction. The evaluation of the profiles of oil yield, TPH and chlorophylls in oils obtained under different co-extraction conditions (water and thyme concentrations) was only possible via the visual observation of each response surface fitted to each data set. The optimal co-extraction conditions were chosen from the information shown in these response surfaces.”

Comment 6: Conclusions: some statements reported in this section are the same reported in the abstract section. Please try to differentiate and avoid to report the same information twice. In this section the authors should try to emphasize the relevance of this work to new findings in this field and the ways in which the new results have advanced the field.

Ans: Thank you for your suggestions. We modified the conclusions as follows:

“Consumer’s attitude towards olive oil is changing. Their search for different flavored oils that match several food pairings is an opportunity for the olive oil sector. Thus, the research on developing adequate technologies to produce these novel oils, presenting adequate sensory properties, improved biological value and shelf-life stability, is of utmost importance for the olive oil industry. Although different techniques can be used, some of them consisting of the incorporation of the flavoring agent in the olive oil may have a deleterious effect on nutritional and shelf-life characteristics. This study showed the feasibility of sustainably producing new high-value products obtained with oils from autochthonous olive cultivars, using natural resources. Olive oils obtained from over ripe olives, presenting slight aroma and low amounts of bioactive phenolic compounds, will be highly valorized by co-extraction with thyme. The optimization of co-extraction and storage studies is a good way to evaluate the main effects of the addition of thyme. The co-processing of ripe olives with T. mastichina, added either in the milling or in the malaxation operations, showed to be a feasible and easy technique to obtain flavored oils with intense thyme sensory notes and without sensory defects that were present in the VOO extracted from the same olives. Thus, under optimized co-processing conditions by RSM, flavored oils presented higher phenolic contents, and higher biologic value, than the non-flavored VOO. The flavored oil obtained by co-processing with T. mastichina addition in the mill showed higher oxidative stability than the VOO and the co-processed oil with thyme addition in the malaxator, even after six-month storage in the dark.”

Best regards,

Suzana Ferreira-Dias

Round 2

Reviewer 3 Report

I believe that the manuscript has been sufficiently improved and warrant publication in the Special Issue of Life, “Advances in Edible Oil Processing”. 

Best regards